# Synergistic Effect of Combination of Various Microbial Hurdles in the Biopreservation of Meat and Meat Products—Systematic Review

**DOI:** 10.3390/foods12071430

**Published:** 2023-03-28

**Authors:** Marcelina Karbowiak, Piotr Szymański, Dorota Zielińska

**Affiliations:** 1Department of Food Gastronomy and Food Hygiene, Institute of Human Nutrition Sciences, Warsaw University of Life Sciences (WULS-SGGW), Nowoursynowska 159C St., (Building No. 32), 02-776 Warsaw, Poland; marcelina_karbowiak@sggw.edu.pl; 2Department of Meat and Fat Technology, Waclaw Dabrowski Institute of Agricultural and Food Biotechnology—State Research Institute, Rakowiecka 36 St., 02-532 Warsaw, Poland

**Keywords:** synergism, compounds, meat safety, shelf life extension, bacterial-derived antimicrobials, decontamination strategies, foodborne pathogens

## Abstract

The control of spoilage microorganisms and foodborne pathogens in meat and meat products is a challenge for food producers, which potentially can be overcome through the combined use of biopreservatives, in the form of a mix of various microbial hurdles. The objective of this work is to systematically review the available knowledge to reveal whether various microbial hurdles applied in combination can pose an effective decontamination strategy for meat and meat products. PubMed, Web of Science, and Scopus were utilized to identify and evaluate studies through February 2023. Search results yielded 45 articles that met the inclusion criteria. The most common meat biopreservatives were combinations of various starter cultures (24 studies), and the use of mixtures of non-starter protective cultures (13 studies). In addition, studies evaluating antimicrobial combinations of bacteriocins with other bacteriocins, BLIS (bacteriocin-like inhibitory substance), non-starter protective cultures, reuterin, and S-layer protein were included in the review (7 studies). In one study, a biopreservative mixture comprised antifungal protein PgAFP and protective cultures. The literature search revealed a positive effect, in most of the included studies, of the combination of various bacterial antimicrobials in inhibiting the growth of pathogenic and spoilage bacteria in meat products. The main advantages of the synergistic effect achieved were: (1) the induction of a stronger antimicrobial effect, (2) the extension of the spectrum of antibacterial action, and (3) the prevention of the regrowth of undesirable microorganisms. Although further research is required in this area, the combination of various microbial hurdles can pose a green and valuable biopreservation approach for maintaining the safety and quality of meat products.

## 1. Introduction

Foodborne pathogens are a threat to international public health and safety. A majority of zoonosis cases reported at the European Union level in 2021 were campylobacteriosis, with 127,840 confirmed cases, followed by salmonellosis, Shiga toxin-producing *Escherichia coli* (STEC) infections, yersiniosis, and listeriosis with 60,050, 6789, 6084 and 2183 confirmed cases, respectively [1]. On the other hand, multistate outbreaks were most frequently caused by *Salmonella* (83 outbreaks, 64%), followed by STEC (29 outbreaks, 22%), and then *Listeria* (18 outbreaks, 14%) [2]. Meat and meat products were the main carriers of these pathogens, in both the European Union and the United States [1,2]. In the United States, during the period 2017–2020, beef consumption was associated with 900 hospitalizations due to *E. coli*, *Salmonella*, or *Listeria* contamination. At the same time, the consumption of contaminated chicken was linked to 679 cases and pork to 126 cases [2]. In the European Union, chicken meat (broilers) is considered to be the most important foodborne source of both campylobacteriosis and salmonellosis, which are the two most frequently reported foodborne human diseases [1]. While meat and meat products are still among the best sources of nutrients for humans due to their high protein content, essential amino acids, vitamin B groups, and minerals [3], their high water activity and nutrient composition also contribute to the development of foodborne pathogens, as well as spoilage microorganisms, including *Pseudomonas*, *Staphylococcus*, *Brochothrix*, some lactic acid bacteria (LAB) species, and different strains of *Enterobacteriaceae* [4,5]. There are also several metabolites secreted by spoilage microorganisms that can negatively impact meat quality and pose health risks, including biogenic amines (histamine, putrescine, cadaverine, spermine, and spermidine), as well as toxins (botulinum and staphylococcus toxin) [6]. Development of these bacteria in meat leads to the reduction of nutritional compounds, primarily proteins, amino acids, and vitamins (mostly B complex vitamins), as well as degradation of lipids, which are more susceptible to oxidation in these environments [4]. Additionally, important sensory characteristics such as color, texture, and odor are altered, making the meat unfit for consumption. 

To ensure the quality and safety of products, and in response to increasing pressure from consumers for safe-to-eat and high-quality clean-label food products, interest in the meat industry is now focused on novel biopreservation strategies. Biopreservation involves using natural or primary and/or secondary metabolites or antimicrobials, from such sources as bacteria, fungi, plants, and animals, as a means to minimize lipid oxidation, reduce color loss, or extend product shelf life [7]. According to this approach, biopreservatives are defined as compounds, derived from natural sources or made from food ingredients, which are capable of preventing or retarding spoilage related to chemical or biological deterioration and combating foodborne pathogens [8]. In practice, it is more common for authors to define the concept of ‘biopreservation’ with reference merely to the use of microorganisms and/or their metabolites in order to extend shelf life and enhance food safety [9]. Lactic acid bacteria (LAB) possess the biggest potential for use in biopreservation, being used as a starter or protective culture due to the wide spectrum of their activity against the development of unfavorable microbiota. Their antagonistic effect is due to different mechanisms, such as competition for nutrients and production of organic acids, hydrogen peroxide, enzymes, antimicrobial peptides, or bacteriocins [8]. Bacteria of the former genus *Lactobacillus*, which has recently been reclassified into twenty-five new species [10], *Bifidobacterium*, and other microorganisms such as *Enterococcus faecium*, *E. faecalis*, *Streptococcus salivarius* ssp. *thermophilus*, *Lactococcus lactis* subsp. *lactis*, *Leuconostoc mesenteroides*, and *Pediococcus acidilactici* are of especial interest in the biopreservation of meat and meat products [11]. The vast majority of LAB and their metabolites have the status ‘Generally Recognized As Safe’ (GRAS) according to the U.S. Food and Drug Administration (FDA) [12]. The European Food Safety Authority (EFSA) has also given the rank of ‘Qualified Presumption of Safety’ (QPS) to many LAB species, included in the genera *Carnobacterium*, *Lactococcus*, *Leuconostoc*, *Oenococcus*, *Pediococcus*, *Streptococcus* and the former *Lactobacillus* genus [13]. To conclude, an ideal biopreservation agent should be able to exhibit specific antimicrobial activity to combat microorganisms that cause a food safety risk and should not cause adverse effects on the human intestinal microbiome.

Alternatively, biopreservatives might also be used in combination with other strategies as part of a hurdle technology approach to decontaminate food. A hurdle concept is the combination of different preservation techniques as a conservation strategy, and has been used successfully in many countries for years [14]. The combined use of multiple preservation methods may include physical (e.g., heat treatment), chemical (chemical preservatives such as EDTA, organic acid, etc.), and biological (microorganisms and their metabolites) factors [15]. A combination of these approaches is rationalized by the belief that synergism may occur by exposing undesired microorganisms to a series of obstacles to their growth and survival. Furthermore, if synergism occurs, preservative doses or technological treatment intensities may be reduced [15]. Using hurdle technology in the meat industry is a successful method of obtaining mildly processed meat products that are both safe and appealing to consumers [16]. However, foodborne disease outbreaks linked to meat products continue to occur despite promising evidence that these interventions result in safer products. Up to this date, several reviews have been published, where an attempt was made to collect available data on meat biopreservation strategies, some belonging to the category of hurdle technology [9,15,17,18,19,20,21,22,23,24]. According to a preliminary literature search, there has not yet been a review of the combination of bacterially derived antimicrobials to achieve a synergistic effect in the decontamination of meat and meat products.

In light of the above, as outbreaks of foodborne diseases related to meat products continue to be reported, and based on the numerous positive reports describing the antimicrobial and potentially synergistic effects of preservatives of microbial origin, the purpose of this study was to review systematically the current knowledge about the combined antimicrobial effect of a variety of microbial hurdles to control the growth of undesirable microorganisms in meat and meat products. 

## 2. Materials and Methods

To the best of the authors’ knowledge, until today there is no published or registered protocol concerning this topic or a similar review of the literature. The review was registered with the PROCEED prospective register of evidence syntheses in the environmental sector (registration number PROCEED-23-00084), available at the following address: https://www.proceedevidence.info/protocol/view?id=84 (accessed on 28 February 2023). Guidelines and Standards for Evidence Synthesis in Environmental Management Version 5.1 were strictly followed in this systematic review [25].

### 2.1. Focus Question

What is the impact of the mixture of bacterial-origin antimicrobials on the growth of foodborne pathogens and spoilage bacteria in the food system? Can this biopreservative effect be accomplished by using specific mechanisms and/or compounds? Is there a way to utilize these combined bacterial antimicrobials in the meat industry?

### 2.2. Protocol and Selection Criteria

The population, intervention, control, outcome (PICO) design methodology was applied in this case [26]. “Population” constituted meat and meat products, “intervention” pertained to the combination of a variety of antimicrobial agents of bacterial origin, “comparator” constituted no combination used, single antimicrobial or other agents, and “outcome” referred to decontamination strategies and effects. Studies were qualified to be included if they met the following criteria: published since 2000, comprising experimental in vitro research, and referring to the synergistic antibacterial effect of diverse microorganisms or microbial origin metabolites applied in combination to enhance food safety or extend the shelf life of contaminated meat and meat products. Studies were excluded when related to animal or in vivo research, to human trials, and when lacking sufficient details on either the antimicrobial agents or their utilization in the meat system. Unpublished or duplicate data, reviews, letters, case reports and editorials were also excluded.

No restrictions were placed regarding the use of control samples, in order to include a wide range of potential studies. To that end, studies without a comparator, e.g., without comparing to antimicrobial agents used separately, were also included.

A combination of microbial hurdles with organic acids and their derivative salts was excluded at the research selection stage as having no certainty as to the microbial origin of the chemicals. It should be noted that lactic acid, citric acid, propionic acid, acetic acid, and other organic acids are chemical compounds that can be produced by microbial activity; however, chemical synthesis and enzymatic production processes are also possible ways to obtain these antimicrobials [27]. Thus, they are typically classified as chemical additives for controlling microorganism growth, improving sensory attributes, and extending the shelf life of various food products, including meat [15]. 

### 2.3. Search Methods

The following databases were searched in February 2023: PubMed, Scopus, and Web of Science, with the search restricted to the English language. Following the methodology of Bramer et al. (2018) [28], keywords were carefully selected and tested in order to provide a broad screening of the available and relevant literature. Search queries were tailored to each database’s structure. The specific search terms combined with the Boolean operators, as well as details on the customization of the query to each database, are provided in the PROCEED protocol. Additionally, titles and abstracts of the literature were screened for duplication across search engines.

### 2.4. Selection of Articles

The title and abstract of each article were independently reviewed by two researchers to ensure they met the eligibility criteria. Full texts of articles were then retrieved, and two researchers independently screened them for inclusion. Disagreements were settled through discussion between the two researchers. The PRISMA diagram [29] was used to guide the review process (Figure 1). Citations were exported to the citation generator.

### 2.5. Risk of Bias Assessment 

The Guidelines for Systematic Review and Evidence Synthesis in Environmental Management Version 5.1 2022 [25], Collaboration for Environmental Evidence Critical Appraisal Tool Version 0.3 (Prototype) was utilized for a critical appraisal of included studies that met the eligibility criteria. This tool is intended for use in environmental management research, including pathogen control, which is one of the research topics of this review. Criteria for avoiding systematic biases which may pose a threat to the study’s internal validity are described in detail in the PROCEED protocol. 

### 2.6. Search for Heterogeneity

Given the high heterogeneity among the studies, meta-analyses could not be performed because similar comparisons could not be made between outcomes that differed between the studies. 

## 3. Results

### 3.1. Study Selection

A total of 1307 papers were identified, 134 in the PubMed database, 452 in Scopus, and 721 in the Web of Sciences. These three lists were compared identifying 452 duplications or triplicates. The title and abstract screening allowed exclusion of 748 articles since they lacked relevance to the topic. Among the 107 articles subjected to full-text review, 82 articles were excluded as not complying with the inclusion criteria. An additional twenty relevant studies were identified by checking the references from included studies or searching the Internet and were added to the final analysis. In total, forty-five unique studies were selected for data analysis. The large number of studies that were identified additionally, rather than through a systematic search, may be because the studies’ main objectives were not to investigate synergism between antimicrobial agents. It is possible that the authors did not use such keywords as “synergism”, “combination” or “additivity”, and the synergistic effect was obtained somewhat accidentally.

The selection strategy employed in the qualitative analysis is summarized in Figure 1. To examine the topic deeply, a content analysis of all articles found was conducted. The data extracted from the studies included in the systematic review are expressed in Table 1, Table 2, Table 3, Table 4, Table 5, Table 6, Table 7 and Table 8. 

### 3.2. Risk of Bias

All forty-five studies included in this review were considered to be moderately biased. A detailed description of the scoring results of individual studies will be provided upon request.

### 3.3. Main Findings

Evaluation and in-depth examination of the articles included in this review showed that the vast majority of interventions were the addition of starter cultures directly to meat products during the fermentation process. Another common mixture of antimicrobial agents was a combination of non-starter protective cultures. In this case, the target meat products were not fermented but fresh, such as minced meat or ready-to-eat products. In both cases, the mix of agents was added fresh or lyophilized, or in the form of the supernatants obtained by centrifugation of bacterial cultures. In other cases, the potential use of secondary metabolites and purified or semipurified biologically active compounds of microbial origin, such as bacteriocins, bacteriocin-like substances, surface layer proteins, and reuterin were assessed in different variations, as a part of the decontamination strategies for meat. Often, combined physical or chemical hurdles technology approaches and active packaging techniques were applied simultaneously.

Therefore, it was decided to use these categories in the rest of the study. To determine whether microbial hurdle technology has an advantage over microbial biopreservatives used alone, the articles included in the review were considered according to their function, namely, whether they were added into the meat as a mixture of starter cultures, or used as bioprotective non-starter cultures. Other combinations of microbial secondary metabolites were considered separately.

The most popular microbial hurdles in the studies included in this review were combined starter cultures applied to meat [30,31,32,33,34,35,36,37,38,39,40,41,42,43,44,45,46,47,48,49,50,51,52,53]. The use of a mixture of non-starter protective cultures to inhibit spoilage bacteria or/and foodborne pathogens was also assessed [54,55,56,57,58,59,60,61,62,63,64,65,66]. Gao et al. (2015) [67] investigated the antimicrobial properties of the non-starter protective culture of *Latilactobacillus sakei* C2 in combination with bacteriocin sakacin C2 prepared from the cell-free supernatant of *L. sakei* C2. In the study of de Souza de Azevedo et al. (2019) [68], nisin and bacteriocin-like inhibitory substance (BLIS) produced by *Pediococcus pentosaceus* ATCC 43,200 were tested together to preserve pork meat samples after artificial contamination by *Lactobacillus sakei* ATCC 15521. Interestingly, in the Delgado et al. study (2018) [69], non-starter protective cultures were combined with protein PgAFP from *Penicillium chrysogenum* in order to investigate their combined antifungal capacity against *A. parasiticus* in dry-fermented sausage. In turn, the biopreservation of meat by using a combination of bacteriocins was evaluated by Castellano et al. (2018) [70] and Vignolo et al. (2000) [71]. In other similar studies, combinations of nisin with bacteriocin-like-substance (BLIS) against *Listeria monocytogenes* [72] and with surface layer protein (SLP) against *Staphylococcus saprophyticus* P2 [73] were appraised. The inhibitory activity of synergetic antimicrobial consortia of, inter alia, reuterin and microcin J25 against *Salmonella enterica* on broiler chicken carcasses was also evaluated [74]. 

## 4. Discussion

### 4.1. Combination of Various Microbial Hurdles against Foodborne Pathogens and Spoilage Microorganisms in Meat and Meat Products

#### 4.1.1. Combination of Starter Cultures

Starter cultures refer to preparations that contain a large number of cells, whether of a single type or a mixture of two or more microorganisms, that are added to food to gain benefits from their metabolism and enzyme activity [75]. Across the meat industry, starter cultures are widely used, primarily in the preparation of sausages. Fermentation is the oldest known method of preserving meat, obtaining a microbiologically stable product with special sensory properties that can be stored for months. Habitually, the fermentation process of meat products has been conducted by the natural microbiota present in the meat [76]. The use of starter cultures in the manufacture of meat products has received considerable attention in recent years. A variety of strains are used in different products based on technological requirements and consumer preferences [77]. In fermented meat products, the LAB are usually facultative anaerobes and belong primarily to the genera *Leuconostoc*, *Pediococcus*, *Lactococcus*, *Enterococcus*, and the former *Lactobacillus*. Among coagulase-negative staphylococci, *Staphylococcus xylosus* and *S. carnosus* are the most commonly used facultative anaerobes in the fermentation of meat products [19]. Yeasts commonly used as meat starters are *Debaryomyces* spp. and *Candida* spp. which can exhibit either an aerobic or a facultatively anaerobic metabolism [78]. The main objective of using the combinations of various strains within a starter culture is to obtain potent synergistic activity in terms of antimicrobial effect enhancement, sensory quality improvement, reduction of usage, avoiding resistance, saving energy, and the extension of meat shelf-life which is beneficial to the meat industry. 

In the included studies, a starter culture composed of at least two LAB strains was used by Lee et al. (2018) [39], Olaoye et al. (2010) [43], Olaoye et al. (2011) [44], Olaoye et al. (2015) [45], Iacumin et al. (2020) [36], and Vatanyoopaisarn et al. (2011) [51] (Table 1). In the study by Iacumin et al. (2020) [36], two different bioprotective mixed cultures consisting of lactic acid bacteria were used against *L. monocytogenes* bacteria that were intentionally inoculated into cooked cubed ham, which was packaged in modified atmosphere packaging (MAP). The use of two types of bioprotective mixed cultures, including Lyocarni Sacco BOX-74 (*Carnobacterium divergens*, *Carnobacterium maltaromaticum*, and *Latilactobacillus sakei*) and Lyocarni BOX-57 (*C. divergens; C. maltaromaticum* and *L. sakei*—bacteriocin producer) eliminated or prevented the development of *L. monocytogenes* in the meat products [36]. According to the authors, the activity of these bioprotective cultures was mainly based on competition at the substrate level and also, in the case of BOX-57, inclusion of a bacteriocin producer. As demonstrated in the experiment, the presence of autochthonous (natural) LAB did not inhibit *L. monocytogenes*, and the mixed three-strain starter cultures did [36].

In the remaining studies concerning the use of a combination of strains in a starter culture, the LAB were combined also with coagulase-negative staphylococci (CNS), which possess different properties. Due to their nitrate-reducing capabilities, coagulase-negative staphylococci contribute to limiting lipid oxidation and producing nitrosomyoglobin, thus promoting the development of the typical red color of meat [79]. *Staphylococcus* species, such as *Staphylococcus xylosus*, also contribute to aroma formation by performing proteolytic and lipolytic activities. According to several recent studies, protein hydrolysis is caused not only by endogenous meat enzymes but also by certain bacteria, including *Staphylococcus* [80,81]. Interestingly, the success of *Staphylococcus* in proteolysis during meat fermentation may depend directly on the presence of lactic acid bacteria [82], which indicates collective interaction between those species. In the study by Chen et al. (2020) [33] on the production of dry fermented sausage, the multi-strain starter of *Staphylococcus xylosus* SX16 and *Lactiplantibacillus plantarum* CMRC6 was compared to the single-strain starter culture of *L. plantarum* CMRC6 and a non-inoculated control. Besides acceleration of acidification and proteolysis during ripening, the mixed culture starter improved the microbiological safety of the meat sausages, dominating the microbial community by suppressing *Enterobacteriaceae* growth which was not detected after 6 days of ripening. In comparison, in sausages inoculated with the single culture starter, the *Enterobacteariaceae* population decreased more slowly and was not detected after 14 days of ripening but no information was given as to whether the difference was significant [33]. 

Similar results of improvement of the safety and biochemical and sensory characteristics of the fermented meat products when using the combination of LAB and CNS strains were obtained by Du et al. (2019) [64], Frece et al. (2014) [35], Pavli et al. (2020) [46], Zhao et al. (2011) [53], Xiao et al. (2020) [52], Bonomo et al. (2011) [30], Casaburi et al. (2017) [31], Mafra et al. (2020) [40], and Najjari et al. (2021) [42] (Table 1). It is worth mentioning that in the study of Pavli et al. (2020) [46], a three-strain starter culture of *Pediococcus pentosaceus*, *Staphylococcus carnosus* and *Lactiplantibacillus plantarum* L125 was compared with a two-strain starter culture of *P. pentosaceus* and *S. carnosus* and applied separately to meat batter. The use of the three-strain starter culture led to equal or better technological characteristics of the produced sausages compared to the control sample. At the same time, the population of staphylococci in meat samples inoculated with the mix of three strains decreased more than in meat samples with the addition of the two-strain starter culture (*p* < 0.05) in the initial stage of fermentation, probably due to the strong competition between LAB cultures. Despite this fact, staphylococci levels were higher in meat samples incorporated with the three-strain starter culture by the end of the storage period (182 days) compared to the meat sample impoverished by the *L. plantarum* strain. Overall, the mix of three-starter cultures was found to possess desirable technological characteristics, indicating its effectiveness for use in fermented sausage manufacturing [46]. In the study by Mafra et al. (2020) [40], the mixed starter culture composed of *Latilactobacillus sakei*, *Staphylococcus xylosus*, and *Staphylococcus carnosus* was used in the fermentation of meat sausages and resulted in improvement of the safety and shelf-life of products. According to the authors, the mixed starter culture presented technological characteristics anticipated for application in the maturation of sausages [40]. 

**Table 1 foods-12-01430-t001:** The use of diverse combinations of starter cultures against different target microorganisms.

Mixture	Meat System	Target Microorganism(s)	Synergism Occurrence	References
Starter cultures of *Latilactobacillus sakei* and *Staphylococcus equorum*	Traditional fermented sausages of Basilicata region	LAB, CNS, *Enterobacteriaceae*, gram-negative bacteria, molds, and yeasts	Yes	Bonomo et al. (2011) [30]
Starter cultures of *Staphylococcus xylosus* CVS11 or FVS21 with *Latilactobacillus curvatus*	Fermented sausages	*Enterobacteriaceae*, Enterococci, molds, yeast, LAB, and *Micrococcaceae*	Yes	Casaburi et al. (2007) [31]
Starter cultures of *Pediococcus acidilactici* (MC184, MS198, or MS200) and *Staphylococcus vitulus* RS34	Traditional Iberian dry-fermented salchichón and chorizo	Various pathogens (*Listeria*, *Salmonella*, *E. coli*, *S. aureus*), *Enterobacteriaceae*, and *Micrococcus*	Yes	Casquete, et al. (2012) [32]
Starter cultures of *Staphylococcus xylosus* SX16 and *Lactiplantibacillus plantarum* CMRC6	Gourmet fermented dry sausage	LAB, CNS, and *Enterobacteriaceae*	Yes, but only concerning *Enterobacteriaceae*	Chen et al. (2020) [33]
Starter cultures of *Pediococcus pentosaceus* and *Staphylococcus xylosus*	Xiangxi sausages	TVC, LAB, *Staphylococcus*, and *Enterobacteriaceae*	No significant difference with the control sample	Du et al. (2019) [34]
Autochthonous starter culture of *Lactiplantibacillus plantarum* 1K and *Staphylococcus carnosus* 4K1	Traditional Croatian dry sausages	*L. monocytogenes*, *Salmonella* ssp., *S. aureus*, *E. coli*, *Enterobacteriaceae*, yeasts, and molds	Yes	Frece et al. (2014) [35]
Two starter cultures of Lyocarni BOX-74 (*Carnobacterium divergens*, *Carnobacterium maltaromaticum*, and *Latilactobacillus sakei*) and Lyocarni BOX-57 (*Carnobacterium divergens*, *Carnobacterium maltaromaticum*, and *Latilactobacillus sakei* bacteriocin producers)	Cooked cubed pork ham	LAB, *L. monocytogenes*, TVC	Yes	Iacumin et al. (2020) [36]
Two starter cultures of the mix of *Pediococcus acidilactici*, *Latilactobacillus curvatus* + *Staphylococcus* *xylosus*, and *Latilactobacillus sakei + Staphylococcus carnosus*	Sucuk, Turkish dry-fermented sausage	*S. aureus*, LAB, *Micrococcus/Staphylococcus*, *Enterobacteriaceae*	Yes	Kaban et al. (2006) [37]
Two starter cultures of the mix of *Staphylococcus xylosus* DD-34, *Pediococcus acidilactici* PA-2 + *Latilactobacillus bavaricus* MI-401, and *S. carnosus* MIII + *Latilactobacillus curvatus* Lb3	Dry sausage	*Escherichia coli* O157:H7, *Listeria monocytogenes*	Yes	Lahti et al. (2001) [38]
Starter cultures of *Lactobacillus* spp., *Leuconostoc* spp., *Lactococcus* spp., *Pediococcus* spp., and *Weissella* spp.	Fermented sausages	TVC, yeast mold, and LAB	Yes, but no significant difference with the commercial LAB starter culture used as a control sample	Lee et al. (2018) [39]
Starter culture of *Latilactobacillus sakei*, *Staphylococcus xylosus*, and *Staphylococcus carnosus*	Meat sausages	*Escherichia coli* ATCC25922, *Salmonella* Enteritidis ATCC13076, *Vibrio parahaemolyticus*, *Staphylococcus aureus* ATCC43300, *Enterococcus faecalis* ATCC29212, and *Listeria monocytogenes* CERELA	Yes	Mafra et al. (2020) [40]
Starter culture of *Pediococcus pentosaceus* and *Staphylococcus carnosus* with co-cultures of *Limosilactobacillus reuteri* and *Bifidobacterium longum*	Dry fermented sausages	*Escherichia coli* O157:H7	Yes	Muthukumarasamy & Holley (2007) [41]
Starter culture of *Latilactobacillus sakei* (23K, BMG 95, or BMG 37) and *Staphylococcus xylosus*	Tunisian dry-fermented sausages	*S. aureus*, *Salmonella* spp., total coliforms, LAB, anaerobic sulphate-reducing bacteria, yeast, and molds	Yes	Najjari et al. (2021) [42]
Starter culture of *Pediococcus pentosaceus* LIV 01 and *P. acidilactici* FLE 01	Sliced fresh beef	*Enterobacteriaceae*, *Staphylococcus*, yeasts, molds, *Listeria monocytogenes*, and *Salmonella* Typhimurium	Yes	Olaoye et al. (2010) [43]
Starter culture of *Pediococcus pentosaceus* GOAT 01 and *Lactiplantibacillus plantarum* GOAT 012	Goat meat	*Enterobacteriaceae*, *Staphylococcus*, yeasts, molds, *Listeria monocytogenes*, and *Salmonella* Typhimurium	Yes, but no significant difference with the control sample concerning yeast and mold counts	Olaoye et al. (2011) [44]
Starter cultures of *Lactococcus lactis* subsp. *lactis* I23 (Llac01) and *Lactococcus lactis* subsp. *hordinae* E91 (Llac02)	Pork meat	*Brochothrix thermosphata*	Yes	Olaoye et al. (2015) [45]
Starter cultures of *Pediococcus pentosaceus* and *Staphylococcus carnosus* with *Lactiplantibacillus plantarum* L125	Traditional Greek dry-fermented sausage	*Pseudomonas* spp., *Brochothrix* spp., *Enterobacteriaceae*, yeasts, molds, and *Listeria monocytogenes*	Yes	Pavli et al. (2020) [46]
Commercial starter culture (FloraCarn) consisting of a mixture of *Pediococcus pentosaceus* and *Staphylococcus xylosus* in combination with a non-traditional meat starter culture of dairy or human intestinal origin	Hungarian salami	*Listeria monocytogenes* and *Escherichia coli* O111	Yes	Pidcock et al. (2002) [47]
Starter culture of *Staphylococcus xylosus* and *Lactiplantibacillus plantarum*	Harbin dry sausage	TVC, LAB, and *Enterobacteriaceae*	Yes, but only concerning *Enterobacteriaceae*	Sun et al. (2016) [48]
Starter culture of *Staphylococcus xylosus* and *Lactiplantibacillus plantarum*	Harbin dry sausage	TVC, LAB, and *Enterobacteriaceae*	Yes	Sun et al. (2019) [49]
Starter cultures of *Limosilactobacillus fermentum* S8 and *Staphylococcus carnosus* ATCC 51365	Canned minced pork meat	TVC, LAB, *Staphylococcus*	Yes	Szymański et al. (2021) [50]
Starter cultures of diverse mix of *Lactiplantibacillus plantarum* CP1-15, *Lactiplantibacillus plantarum* CP2-11 and *Pediococcus acidilactici* CP7-3	Thai fermented sausage “Sai-Krok-Prew”	LAB, *E. coli*, *Salmonella*, total staphylococci, and *S. aureus*	Yes	Vatanyoopaisarn et al. (2011) [51]
Starter culture of *Lactiplantibacillus plantarum* R2 and *Staphylococcus xylosus* A2	Chinese dry fermented sausages	TVC, LAB, and *Staphylococcus* spp.	Yes	Xiao et al. (2020) [52]
Starter culture of *Lactiplantibacillus pentosus*, *Pediococcus pentosaceus* and *Staphylococcus carnosus*	Mutton sausages	LAB, TVC, micrococci–staphylococci	Yes	Zhao et al. (2011) [53]

Nevertheless, the reason for using synergistic bacterial cultures may be wider than obtaining antimicrobial action. Among the many hazards associated with meat products, nitrosamines, biogenic amines (BAs), polycyclic aromatic hydrocarbons (PAHs), and mycotoxins are among the most significant [48,49,50].

#### 4.1.2. Combination of Non-Starter Protective Cultures

A natural preservation technique for extending the shelf life of fresh meat by inhibiting spoilage and/or pathogenic bacteria is the use of protective cultures. It is worth emphasizing that protective cultures do not change the technological and sensory properties of food [66]. Usually, the bacterial species that serve as protective cultures have long been used as starter cultures for fermented meat products and have the GRAS and/or QPS status, including not only lactic acid bacteria (e.g., lactobacilli, streptococci, enterococci, lactococci) and bifidobacteria but also *E. coli* and species of *Bacillus*, yeasts such as *Saccharomyces*, and molds such as *Aspergillus* [78]. The protective effect may result from several mechanisms, including the production of bacteriocins and other antagonistic molecules (e.g., organic acids, hydrogen peroxide, and enzymes) and the competition for nutrients. There is also speculation that mixtures of protective cultures may be more effective than single isolates [56]. 

The findings from most studies suggest that combinations of protective cultures have the potential to control the growth of foodborne pathogens [54,55,57,58] and spoilage bacteria [60,61,65,66], both of them [56,59], or EPS-producers [63,64] to extend the shelf-life of fresh meat products (Table 2). 

In the study of Xu et al. (2023) [66], the combination of two protective cultures suppressed the growth of spoilage bacteria in premium (lean) beef mince, whereas this inhibitory effect was not observed in standard (higher fat) beef mince [66]. The activity of antimicrobial compounds produced by protective cultures can be reduced or lost due to interaction with fat because some of these compounds contain lipophilic or hydrophobic portions which interact with the cell membranes of bacteria [66]. 

According to Xu et al. (2021) [65], the greatest antimicrobial action was observed when *L. sakei* was present in mixtures with either *S. carnosus*, or *S. xylosus*. The stronger inhibitory effect induced by combined cultures may be attributed to the synergistic effects between the staphylococci and *L. sakei* [65]. However, in the Gargi et al. (2021) [58] study, the synergistic effect of protective bacterial cultures was not obtained. According to this study [58], marination containing a single culture of *L. casei* and unripe grape and onion juices was the most effective against pathogens inoculated on meat surfaces and also met consumer expectations at a high level, in terms of flavor properties. The authors explained this by the effects of specific metabolites produced by the *L. casei* strain during the marination process [58]. The results of this study are in line with [61], where the combination of supernatants did not exceed the antimicrobial effectiveness of individual supernatants [61]. In turn, Castellano et al. (2011) [55] reported the varied effectiveness of protective cultures of *L. curvatus* CRL705 and *L. lactis* CRL1109 in combination with Na_2_EDTA on frozen ground-beef patties contaminated with *Escherichia coli* O157:H7. The bioprotective cultures failed to inhibit *E coli* or coliforms in the absence of Na_2_EDTA, with the pathogen reaching similar counts as with control samples [55]. This inhibitory effect may be attributed to the chelating activity of Na_2_EDTA which binds trace metals valuable for fastidious LAB growth while in the presence of *E. coli*. As compared with substrate metallic trace elements, the chelator exhibits a higher affinity for the outer membrane ions of Gram-negative strains [55]. In the study of Chaillou et al. (2014) [56], a *L. sakei* cocktail of three strains showed an effect against *S*. Typhimurium and *E. coli* under vacuum or modified atmosphere packaging. Real-time quantitative PCR showed that the three inoculated *L. sakei* strains had different growth patterns and that the association of these three strains indeed impaired the growth of pathogens [56]. 

**Table 2 foods-12-01430-t002:** The use of diverse combinations of non-starter protective cultures against different target microorganisms.

Mixture	Meat System	Target Microorganism(s)	Synergism Occurrence	References
Supernatants from *Pediococcus acidilactici*, *Lacticaseibacillus casei*, and *Lacticaseibacillus paracasei*	Frankfurters and cooked ham	*L. monocytogenes*	No significant difference with the control sample	Amézquita & Brashears (2002) [54]
Cultures of bacteriocin-producing strains of *Latilactobacillus curvatus* CRL705 and *Lactococcus lactis* CRL1109	Frozen ground-beef patties	*Escherichia coli* O157:H7	Yes	Castellano et al. (2011) [55]
Six strain combinations containing three different strains of *Latilactobacillus sakei*	Vacuum-packed or modified atmosphere-packed ground beef	*Salmonella enterica* Typhimurium, *Escherichia coli* O157:H7, and *Brochothrix thermosphacta*	The growth of indicator strains was variable and depended on both the storage conditions and the amount of indigenous microbiota	Chaillou et al. (2014) [56]
Neutralized cell-free supernatants from *Latilactobacillus sakei* CWBI-B1365 and *Latilactobacillus curvatus* CWBI-B28	Raw beef and poultry meat	*Listeria monocytogenes*	Yes	Dortu et al. (2008) [57]
Cultures of *Lactobacillus acidophilus* LA5, *Lacticaseibacillus casei* 01 and *Lacticaseibacillus rhamnosus* HN001	Marinated meat	*L. monocytogenes*, *E. coli* O157:H7, and *S*. Typhimurium	No	Gargi et al. (2021) [58]
Postbiotics from *Latilactobacillus sakei* and *Staphylococcus xylosus*	Chicken drumsticks	*Listeria monocytogenes*, *Salmonella* Typhimurium, TVC, psychrotrophic bacteria, and LAB	Yes	Incili et al. (2022) [59]
*Latilactobacillus sakei* 27, 44 and 63 strains	Lamb meat	General anaerobic bacteria	Yes	Jones et al. (2010) [60]
*Latilactobacillus sakei* CECT 4808, and *Latilactobacillus curvatus* CECT 904T	Vacuum-packaged sliced beef	*Enterobacteriaceae*, *Pseudomonas* spp., *Brochothrix thermosphacta*, yeasts and molds, and LAB	No	Katikou et al. (2005) [61]
Three LAB strains (Lactiguard^®^)—La51 (*Ligilactobacillus*), M35 (*Lactobacillus amylovorus*), and D3 (*Pediococcus acidilactici*) in combination with their CFS	Frankfurters	*Listeria monocytogenes*	Yes	Koo et al. (2012) [62]
Supernatants from *Lactobacillus acidophilus* CRL641 and *Latilactobacillus curvatus* CRL705	Bovine meat discs	*Latilactobacillus sakei* CRL1407 (exopolysaccharide producer)	Yes	Segli et al. (2021a) [63]
Supernatants from *Lactobacillus acidophilus* CRL641 and *Latilactobacillus curvatus* CRL705	Bovine fresh lean meat	*Latilactobacillus sakei* CRL1407 (exopolysaccharide producer)	No	Segli et al. (2021b) [64]
*Latilactobacillus sakei*, *Pediococcus pentosaceus*, *Staphylococcus xylosus*, and *Staphylococcus carnosus* in various combinations	Lamb meat	*Brochothrix thermosphacta*, *Pseudomonas* spp., and *Enterobacteriaceae*	Yes	Xu et al. (2021) [65]
Mixed culture containing *Staphylococcus carnosus* and *Latilactobacillus sakei*	Beef mince	Aerobic counts, LAB, *Enterobacteriaceae*, *Pseudomonas* spp., *Brochothrix thermosphacta*	Yes	Xu et al. (2023) [66]

Finally, Jones et al. (2010) [60] inoculated fresh commercial lamb meat with a cocktail of three *L. sakei* strains and proved that the protective culture is capable of developing into dominant components of stored meat bacterial flora under standard commercial chilled storage conditions without reducing sensory acceptability [60].

Currently, bacteria are used in food in two different forms: (1) the incorporation of living bacteria into the matrix of foods, as well as (2) the use of purified antimicrobials obtained from major strains. There is no doubt that both methods have their strengths and weaknesses, depending upon the type of food and the nature and characteristics of each strain of bacterium [83]. Viable microorganisms cells can be used directly in food, but there are limitations since their growth and survival are influenced by external factors such as food type, temperature and pH. On the other hand, there are some limitations to the use of bacteriocins. Proteins and lipids appear to interact with bacteriocins in fresh meat and meat products. Bacteriocins and bacteriocin-like inhibitory substances (BLIS) may be also inhibited by proteolytic enzymes [84]. There is also a high risk of contamination of food by Gram-negative bacteria, which are naturally resistant to the action of bacteriocins produced by Gram-positive bacteria, that are widely explored in foods [85]. On the other hand, bacteriocins are costly substances due to the expensive and laborious purification process [86]. The application of postbiotics can be an alternative to reap the benefits of all these natural bacterial antimicrobials. The word “postbiotics” is used to refer to the “preparation of inanimate microorganisms and/or their components that confers a health benefit on the host” [87]. Practically, postbiotics are bioactive soluble factors, products, or metabolites produced by food-grade microorganisms during their growth and fermentation in complex microbiological cultures. On that account, they are sometimes called cell-free supernatant (CFS). Postbiotics are credited with exerting health benefits to the host equivalent to probiotics [88]; however, in this context, there has been discussion of the latest applications of postbiotics in terms of food safety. The inhibitory effects exhibited by postbiotics include producing antimicrobial agents that fall into the category of low molecular weight compounds (i.e., hydrogen peroxide, carbon dioxide, and di-acetylene) or high molecular weight compounds (i.e., bacteriocins and bacteriocins-like substances) [89]. According to recent studies, postbiotics derived from lactic acid bacteria exhibit inhibitory properties toward different groups of microorganisms including *Listeria monocytogenes*, *Staphylococcus aureus*, *Escherichia coli*, *Salmonella* spp., *Yersinia* spp., *Aeromonas* spp., *Bacillus* spp., viruses, yeast, and molds [83]. Nevertheless, most of those studies have been conducted in vitro, whereas studies on food matrices are relatively rare [84]. On the other hand, data on the use of postbiotic cocktail combinations for the biopreservation of food, including meat and meat products, are limited. Six studies, identified in the following review, aimed to apply antagonistic postbiotic combinations in a meat model to inhibit meat potential spoilage or pathogens [54,57,59,61,63,64] (Table 2). In the study of Segli et al. (2021a) [63], the effect of postbiotic extracts from *L. acidophilus* CRL641 (BE-1) and *L. curvatus* CRL705 (BE-2) against the exopolysaccharide producer *L. sakei* CRL1407 in vacuum-packaged meat discs was evaluated. The postbiotics mixture led to a greater growth reduction of 3.31 log CFU/g compared to the control, while the supernatants applied individually reduced bacterial growth by 2.11 and 1.35 log CFU/g, respectively [63]. Interestingly, the inhibition of *L. sakei* CRL1407 was significantly higher even at lower concentrations of each supernatant (MIC) in the postbiotics mixture than those used on postbiotics added separately (2-fold MIC) [63]. Another study by Segli et al. (2021b) [64] was conducted under aerobic conditions at 4 and 10 °C using the same strains from which the postbiotics were obtained and the same spoilage strain contaminating the meat. It was shown that the postbiotics from *L. acidophilus* completely inhibited the spoilage strain, whereas the postbiotics from *L. curvatus* CRL705 and a combination of both extracts exhibited a bacteriostatic effect [64]. Thus, a lack of synergistic interaction was found, since there was a weaker inhibitory effect of the postbiotics mixture compared to the supernatant obtained from *L. acidophilus* CRL641 used individually [64]. Similarly, in Amézquita and Brashears (2002) [54] study, an antilisterial effect of the combined supernatants of three LAB strains (*Pediococcus acidilactici*, *L. casei*, and *L. paracasei*) was observed in commercial frankfurters and cooked ham under refrigeration (*p* > 0.05).

In a study by Incili et al. (2022) [59], the antimicrobial effect on the meat matrix of a combination of postbiotics (mix of *L. sakei* and *S. xylosus*) was compared with postbiotics obtained from *Pediococcus acidilactici*, instead of comparing to single postbiotics. The combination of postbiotics was characterized by the lowest ability to reduce *S*. Typhimurium relative to all tested antimicrobial agents (*p* < 0.05) [59]. Throughout the storage period, *L. monocytogenes* counts in the postbiotics mixture and *P. acidilactici* postbiotics samples were found lower than the control, while the difference from the control was found insignificant (*p* > 0.05) in both cases [59]. In turn, the study by Dortu et al. (2008) [57] obtained different results for *L. monocytogenes* growth inhibition depending on the meat matrix tested. Neutralized cell-free supernatants from *L. sakei* and *L. curvatus* applied either separately or in combination exhibited antilisterial activity in raw beef, whereas, in poultry meat, the inhibition of *L. monocytogenes* could only be achieved by a combined application of these bacteriocin-producing strains, probably due to the activity of proteases produced by proteolytic spoilage strains present in chicken meat [57].

An interesting result was obtained by Koo et al. (2012) [62], who evaluated the antilisterial activity of a combination of three LAB strains (*L. animalis*, *L. amylovorus*, and *P. acidilactici*) in ready-to-eat (RTE) meat. In addition, an antimicrobial preparation consisting of a mixture of the three strains was combined with their cell-free extract [62]. The combination of the LAB was inhibitory to *L. monocytogenes* inoculated onto frankfurters (0.6 log reduction compared to *L. monocytogenes* control after eight weeks of refrigerated storage), and when postbiotics were added to the LAB preparation even more inhibition was obtained (1.2 log reduction compared with *L. monocytogenes*) [62]. The authors attribute the increased antimicrobial activity after the addition of postbiotics to the supernatant content of various biologically active substances that had a synergistic effect when used in combination (potentially, the effect could be due to a bacteriocin—pediocin, secreted by *P. acidilactici*) [62].

#### 4.1.3. Combination of a Variety of Secondary Microbial Metabolites

##### Combination of Bacteriocins 

One of the most widely described biopreservation methods involving bioprotection focuses on bacteriocins. The term “bacteriocins” refers to ribosomally synthesized antimicrobial peptides produced by bacteria and capable of exhibiting a narrow spectrum of activity (targeting members of the same species), while others display a broader spectrum of activity (targeting other species and genera) [90]. Bacteriocins can inhibit the growth of foodborne pathogens or spoilage organisms, and some of these compounds may serve as biopreservatives in the food industry. The popularity of bacteriocins in the food sector is mainly due to (1) their inhibitory properties against pathogenic bacteria *C. perfringens*, *E. coli*, *Salmonella enteritidis*, *B. cereus*, and *L. monocytogenes* [91]; (2) non-resistance of most foodborne pathogens and spoilage microorganisms to bacteriocins; (3) safety—since most bacteriocins can be degraded by enzymes in the human body, they are safe for the human microbiome, and they also inhibit only sensitive bacteria while leaving beneficial bacteria intact [92]; (4) lack of effect on the nutritional and sensory properties of food. 

However, there have only been a limited number of bacteriocins approved for commercial use out of those discovered and preserved in the laboratory: (1) nisin A produced by *Lactococcus lactis*—the most extensively studied bacteriocin and the only product approved as an additive to food by regulatory agencies (WHO, FDA, EFSA); (2) pediocin PA-1 produced by *Pediococcus acidilactici*—FDA-approved bacteriocin-containing metabolites; (3) starter/protective cultures containing a mix of bacteriocin-producing strains—they can improve both flavor and safety, providing fermentation and preservation simultaneously [93]. There are some difficulties in the wide dissemination of bacteriocins in food technology. (1) Firstly, it is known that some bacteriocins have an antibacterial spectrum that is relatively narrow; (2) cytolysin, a cytotoxic compound produced by some strains of *Enterococcus faecalis*, is associated with acutely fatal outcomes in humans [94] and Gram-negative bacteriocins may also produce some endotoxins that can have a deleterious effect, thus requiring rigorous purification protocols; (3) there may be interactions between bacteriocins and other components of the food matrix (its pH, enzymes, etc.); (4) there are some reports that strains of *L. monocytogenes* and *S. aureus* could become nisin-, plantaricin C19- and sakacin A-resistant [95]; (5) finally, obtaining bacteriocins is associated with very high costs.

There has been considerable interest in the hurdle technology approach, however, involving use of bacteriocins to expand the antibacterial spectrum, reduce usage, enhance the antibacterial effect, and avoid resistance. A synergistic effect between bacteriocins, alone or in combination with other microbial hurdle technologies, may facilitate the adjustment of bacteriocins to maximize the viability loss of the target bacteria and also minimize the development of resistance. Among the microbial hurdles associated with the use of bacteriocins are combinations with (1) other bacteriocins [70,71] (Table 3), (2) BLIS (bacteriocin-like-substance) [72] (Table 4), (3) non-starter protective cultures [67,68] (Table 5), (4) reuterin [74] (Table 6) and (5) the surface layer protein [73] (Table 7). 

The lantibiotic nisin has been the subject of several antimicrobial combination studies aimed at targeting *L. monocytogenes*, often connected with the consumption of ready-to-eat (RTE) meat products; this notorious foodborne pathogen causes listeriosis and has the potential to cause opportunistic infections, which may result in meningitis or sepsis in severe cases [96]. It was found that a combination of more than one LAB bacteriocin may be more effective in preventing the spontaneous emergence of bacteriocin-resistant *Listeria* in fresh lean meat than when used individually [71]. A significant antilisterial effect was observed when bacteriocins from among lactocin 705, enterocin CRL35 (both in the concentration of 17,000 AU/mL) and 5000 IU/mL of nisin were combined in pairs; maximal inhibition was reached when nisin was involved [71]. Moreover, in a study by Castellano et al. (2018) [70], it was shown that a significant reduction in *L. monocytogenes* growth was observed when antimicrobials were combined in the dipping solution. A combination of semi-purified bacteriocins (*L. curvatus* CRL705 (533 AU/mL) + *L. sakei* CRL1862 (266 AU/mL)), organic acids, and nisin (2.500 IU/mL) has a 100% of anti-listeria effect on frankfurters, from day 6 to the end of the storage [70]. The authors suggest that it is more effective to prevent *Listeria* in foods by using a mixture of bacteriocins that belong to different classes, such as nisin, lactocin, and enterocin, which are, respectively, lantibiotics (class I), a two-peptide bacteriocin (class IIB), and pediocin-, lactocin- or sakacin-like (class IIA). The effectiveness of separately used bacteriocins may be weakened by many factors acting simultaneously (adsorption on meat components and/or degradation by proteolytic enzymes) [71]. In contrast, the mixture would probably be bactericidal to more bacterial cells since cells resistant to one bacteriocin would be killed by another [70]. 

**Table 3 foods-12-01430-t003:** The use of diverse combinations of bacteriocins against different target microorganisms.

Mixture	Meat System	Target Microorganism(s)	Synergism Occurrence	References
Bacteriocins from *Latilactobacillus curvatus* and *Latilactobacillus sakei*, in combination with nisin	Frankfurters	*L. monocytogenes* and psychrophilic microbiota	Yes	Castellano et al. (2018) [70]
Bacteriocins nisin, lactocin 705, and enterocin CRL35 in combinations	Fresh lean meat	*Listeria monocytogenes* FBUNT	Yes	Vignolo et al. (2000) [71]

By combining bacteriocins, the antimicrobial effect is enhanced and reduced dosage is practicable, lowering the cost of food preservation. Moreover, the bacteriocins will produce synergistic effects if they act in different modes.

##### Combination of BLIS with Bacteriocin

All bacteria can produce antimicrobial peptides (AMPs), but not all of these peptides are defined as bacteriocins. The non-ribosomally synthesized AMPs are known as BLIS, bacteriocin-like inhibitory substances. BLIS are also defined as antimicrobial peptides that have not been fully characterized in terms of their amino acid sequences and biochemical properties [97]. BLIS have the potential for use as biopreservatives, acting as antagonistic substances, with bactericidal or bacteriostatic potential against Gram-positive and/or Gram-negative bacteria, as well as being innocuous to the producer strain [97]. 

Sant’Anna et al. (2013) [72] (Table 4) tested the ability of BLIS produced by *Bacillus* sp. strain P34 in combination with nisin to control *L. monocytogenes* in chicken sausages. Combining P34 with nisin (64 AU/g BLIS P34 + 6.5 μg/g nisin) reduced viable counts of *L. monocytogenes* more effectively than P34 alone in an amount of 128 AU/g [72]. The synergistic antimicrobial activity of the bacteriocin and BLIS is particularly important in the context that the activity of bacteriocins can be reduced in a fatty environment, including meat. The use of bacteriocins in the food matrix is ineffective with increasing fat content [98]. Bacteriocins usually contain a high concentration of hydrophobic amino acids [99]. In this regard, binding by bacteriocins to charged and hydrophobic macromolecules in food is a significant disadvantage when they are to be used as food biopreservatives [72]. 

**Table 4 foods-12-01430-t004:** The use of a combination of bacteriocin with BLIS against a target microorganism.

Mixture	Meat System	Target Microorganism(s)	Synergism Occurrence	References
Bacteriocin-like substance (BLIS) from *Bacillus* sp. strain P34 and nisin	Chicken sausage	*Listeria monocytogenes*	Yes	Sant’Anna et al. (2013) [72]

##### Combination of Non-Starter Protective Cultures with Bacteriocins 

Based on reports of a synergistic effect of antimicrobial substances when used in combination, in two studies it was decided to check whether the antibacterial effect would be enhanced when bacteriocins were combined with non-starter protective cultures in the form of a cell-free supernatant [67,68] (Table 5). 

**Table 5 foods-12-01430-t005:** The use of diverse combinations of bacteriocins with non-starter protective cultures against different target microorganisms.

Mixture	Meat System	Target Microorganism(s)	Synergism Occurrence	References
Nisin with neutralized cell-free supernatant obtained from *Pediococcus pentosaceus* ATCC 43,200	Pork meat	*Latilactobacillus sakei* ATCC 15521	No	de Souza de Azevedo et al. (2019) [68]
*Latilactobacillus sakei* C2 and sakacin C2	Sliced cooked pork ham	*L. monocytogenes* CMCC 54002	Yes	Gao et al. (2015) [67]

The combination of nisin with the neutralized cell-free supernatant, obtained from *Pediococcus pentosaceus* ATCC 43,200 culture at the same concentration (50%, *w*/*v*), was not superior to nisin alone in reducing pathogenic bacteria in a meat matrix, probably due to a negative interaction that may have occurred between them as a result of the environment [68]. The effectiveness of bacteriocins could be influenced by environmental factors such as pH and temperature, interactions with food components, preparation, inactivation, or uneven distribution of bacteriocin in the medium (e.g., agar medium, liquid medium, food) [100]. In addition, there are considerable differences in sensitivity to bacteriocins among Gram-positive bacteria, and the extent of inhibition appears to vary according to the species, genus, and strain type [101]. All this could have contributed to the lack of synergistic effect observed in this study [68].

On the other hand, Gao et al. (2015) [67] investigated the effect of cell-free supernatant from *L. sakei* C2 (0.2 mL of culture) and partially-purified bacteriocin sakacin C2 (concentration of 640 AU/mL), individually or in combination, on the growth of *L. monocytogenes* during the storage of sliced cooked ham at refrigeration temperature for 60 days. When *L. sakei* C2 and sakacin C2 were used in combination, the entire elimination of the cells of *L. monocytogenes* was observed at 30 days of storage and the mixture had no negative effect on the quality of the meat product [67]. 

##### Combination of Reuterin with Bacteriocin

The antimicrobial compound reuterin (*β*-hydroxypropionaldehyde) is produced by certain strains of *Limosilactobacillus reuteri* during the anaerobic fermentation of glycerol. It exhibits a wide spectrum of antimicrobial activity against foodborne pathogens and spoilage bacteria. There have been several studies that demonstrate the antimicrobial properties of reuterin solutions against Gram-positive and Gram-negative bacteria, yeasts, and molds [102]. Reuterin is thought to cause depletion of free thiol groups in glutathione (GSH), proteins, and enzymes, resulting in an imbalance of the cellular redox status; in turn, this leads to the death of bacterial cells [103]. The efficacy of mixtures of antimicrobial compounds, namely reuterin (2 mM), microcin J25 (0.03 μM), and additionally lactic acid for reducing the viability of *Salmonella enterica* serovar Enteritidis and total aerobes on broiler chicken carcasses were evaluated [74] (Table 6). The results of in vitro study indicated that the combination of reuterin + microcin J25 was synergic, making these compounds effective at four times lower concentrations than those used alone. The mixture sprayed onto chilled chicken carcasses reduced *Salmonella* spp. counts by 0.83 CFU/g (*p* < 0.05) and applied as a post-chill spray, could contribute to food safety by decreasing *Salmonella* counts on chicken carcasses while limiting the number of single metabolites used [74].

**Table 6 foods-12-01430-t006:** The use of a combination of bacteriocin and BLIS against different target microorganisms.

Mixture	Meat System	Target Microorganism(s)	Synergism Occurrence	References
Reuterin (produced by *Limosilactobacillus reuteri*) and microcin J25 (produced by *E. coli* MC4100)	Chicken carcasses	The mixture of *Salmonella* Enteritidis, *Salmonella* Heidelberg, *Salmonella* Newport, and TVC	Yes	Zhang et al. (2021) [74]

##### Combination of the Surface Layer Protein (S-Layer Protein/SLP) with Bacteriocin

Surface layer proteins belong to the group of microbial secondary metabolites. The S-layer proteins are monomolecular crystalline arrays composed of a single homogeneous protein or glycoprotein that ranges in size from 40 kDa to 200 kDa [104]. Lactic acid bacteria S-layer proteins have poorly understood biological functions, but their presence can be associated with probiotic-relevant properties such as promoting bacterial adhesion to host cells or extracellular matrix proteins [105]. It has also been demonstrated that S-layer proteins inhibit the growth of some pathogens. According to Meng et al. (2015) [106], the SLP derived from *Lactobacillus acidophilus* damaged the cell walls and membranes of *Escherichia coli*, inhibiting their growth. A study by Sun et al. (2017) [73] aimed to determine the effect of the combination of SLP and nisin against the foodborne spoilage bacterium *S. saprophyticus* P2 in chicken meat and to clarify how SLP acted synergistically with nisin. It has been demonstrated that SLP alone had a controlling effect on total viable counts in the meat model, although to a lesser extent than nisin [73]. It was found that SLP significantly enhanced nisin’s ability to control microbial growth. By using nisin + SLP and 0.5 nisin + SLP combinations, the microbiological shelf life was extended by six days [73]. As for the mechanism of action, the mode of action of SLP/nisin applied individually was different from the combination of SLP and nisin. The authors reported that, in the combination condition, SLP bonded to the cell wall through electrostatic interactions, which not only reduced cell wall integrity in vegetative bacteria and intensified the access of nisin to form a stable pore on the cell membrane but also influenced plasma membrane permeabilization [73]. As a result of these actions, cell content was released fiercely from the damaged cells, preventing them from producing energy and ultimately leading to the death of the damaged cells. The cell lysing effect was not observed when SLP or nisin was used separately [73]. Therefore, in the food industry, nisin + SLP may prove to be a new antibacterial combination that can be utilized to preserve food (Table 7). 

**Table 7 foods-12-01430-t007:** The use of a combination of bacteriocin and surface layer protein against a target microorganism.

Mixture	Meat System	Target Microorganism(s)	Synergism Occurrence	References
Surface layer protein isolated from *Lactobacillus crispatus* K313 (SLP) and nisin	Minced chicken meat	*Staphylococcus saprophyticus* P2	Yes	Sun et al. (2017) [73]

##### Combination of PgAFP with Protective Cultures

Certain molds produce proteins which inhibit other molds and some yeasts, while the activity against prokaryotes is quite limited [69]. PgAFP protein from *Penicillium chrysogenum* is a molecule that belongs to the group of small, cysteine-rich, and basic proteins with fungistatic activity [107]. Molds that produce aflatoxins can grow on food, especially fermented ones. The use of antifungal proteins produced by molds represents a novel and promising biopreservation strategy. The search identified one study that investigated the antifungal capability of PgAFP in combination with protective cultures of *Debaryomyces hansenii* and/or *Pediococcus acidilactici* against *A. parasiticus* in, inter alia, dry-fermented sausage [69] (Table 8). The combination of these two agents was chosen because the presence of calcium in the environment abolishes the inhibitory effect of PgAFP on certain *Aspergillus* spp. [69,108]. The effect of calcium on PgAFP fungal inhibition may be counteracted by lactic acid bacteria. At the same time, combining this protein with protective cultures may maximize its antifungal effect. PgAFP and *D. hansenii* were found to effectively inhibit *A. parasiticus* growth and aflatoxin production in sliced dry-fermented sausages that had been ripened for up to 15 days [69]. *A. parasiticus* growth or mycotoxin production was not substantially inhibited by the addition of *P acidilactici* as an additional protective culture in meat products. The contribution of *P. acidilactici* thus appears to be insignificant, probably due to the poor ability of the bacteria to grow on dry-fermented sausage. The authors suggest that the highest inhibition reached by Pg + Dh treatments can be attributed to the combined effect of the different mechanisms of action. A consequence of ROS (reactive oxygen species) is the induction of permeability, the loss of membrane integrity, and the induction of apoptosis by PgAFP [69]. In contrast, the inhibitory effect of *D. hansenii* is attributed to its volatile compounds and competition for nutrients and space [109]. The results of this study indicate that PgAFP in combination with *D. hansenii* can successfully control the aflatoxigenic population in dry fermented foods, most likely due to complementary mechanisms of action and overcoming the limitations associated with the inhibitory effect of the calcium-rich environment on the antimicrobial activity of the PgAFP protein.

**Table 8 foods-12-01430-t008:** The use of a combination of PgAFP with protective cultures against different target microorganisms.

Mixture	Meat System	Target Microorganism(s)	Synergism Occurrence	References
Small, basic, cysteine-rich antifungal protein PgAFP from *Penicillium chrysogenum* combined with *Debaryomyces hansenii* and/or *Pediococcus acidilactic*	Dry-fermented sausage	Mold, yeast, LAB, aflatoxin B1, and aflatoxin G1	Yes	Delgado et al. (2018) [69]

### 4.2. Mode of Synergistic Action 

The studies published over the past two decades were reviewed in order to gain a more comprehensive understanding of whether there are advantages of combining microbial hurdles in the biopreservation of meat and meat products. It was decided to search the literature systematically, although this is not a popular method for environmental research. A systematic review can deliver a clear and comprehensive overview of available evidence. As is well known, in vitro results do not necessarily predict the success of various combinations applied in a food matrix (in situ), therefore it was also decided to discuss the combination of microbial hurdles as combined antimicrobial agents in a meat matrix extensively. As opposed to in vitro assays, which are much simpler, a food matrix is an environment in which a number of microbial populations interact and influence the overall structure of the community. Including studies based on the experiments designed to be as close as possible to real industrial conditions seems to be the most appropriate.

The synergistic effect of two antimicrobial agents can be measured using the fractional inhibitory concentration (FIC) index. The FIC is determined by dividing each agent’s minimum inhibitory concentration (MIC) when used in combination with each agent’s MIC when used alone. MIC is the lowest concentration of an antibacterial agent expressed in mg/L (μg/mL) which, under strictly controlled in vitro conditions, completely prevents visible growth of the test strain of an organism [110]. The fractional inhibitory concentration (FIC) index is commonly used to determine the interaction of the two agents in combination, ranging from 0.5 to 4. The parameter may indicate: (i) full synergy (FIC ≤ 0.5), (ii) partial synergy (0.5 ≤ FIC ≤ 0.75), (iii) additive effects (0.75 ≤ FIC ≤ 1.0), (iv) indifferent effects (1.0 ≤ FIC ≤ 2.0) and (v) antagonistic effects (FIC ≥ 2.0) [111]. Systematic review can highlight methodological concerns in research studies that can be used to improve future work in the topic area [112]. In this case, attention should be drawn to the fact that the possibility of synergistic action should be investigated in each included study. In only one [74] of the forty-five studies found in this review, was there an attempt to establish an FIC index. The literature describes a variety of methods for assessing antimicrobial synergy in laboratory conditions. Checkerboard assays and e-tests to evaluate synergy are some examples of these tests [113]. As mentioned above, using the checkerboard method, it is possible to determine the fractional inhibitory concentration (FIC) index. There is one primary disadvantage to the checkerboard assay, which is that it can only examine two antimicrobials simultaneously. On the other hand, the multiple combination bactericidal test (MCBT) allows the evaluation of up to four antimicrobial combinations at a time [113]. It is suspected that the lack of an appropriate parameter to examine this synergism in most of the included studies may be due to the fact that the synergism may have arisen completely by chance because it was not the original intention of the authors. In addition, methods that study synergism, such as FIC or MCBT, are not widespread among studies relating to the use of antimicrobial agents of bacterial origin in the food system. These methods are found more frequently in the study of drugs, especially antibiotics [114]. 

In the included studies, the effect of preventing the development of foodborne pathogens and spoilage bacteria was obtained in the majority of the cases. Thus, the results indicate that microbial hurdles in combination may find application in the meat industry as one of the biopreservation techniques. However, the combination of microbial hurdles was not always compared with these agents used separately. Thus, it is difficult to say unequivocally whether the synergism effect has occurred, and to what extent. According to [30,31,32,33,35,36,37,38,39,40,41,42,43,44,45,46,47,48,49,50,51,52,53,55,57,59,60,62,63,65,66,67,69,70,71,72,73,74], the combination of microbial hurdles predominantly enhanced the antimicrobial effect. However, there is a need for further investigation of the mechanism that caused those combinations. According to Liu et al. (2022), the efficiency of bacteriocins, as well as other antimicrobials, can be reduced by the action of the real food matrix complex, which in turn protects foodborne pathogens and spoilage bacteria [15]. Understanding the interactions and mechanisms between microbial hurdles could help find more precise and optimal approaches for overcoming these challenges. Interestingly, no antagonistic effect was observed in any of the included studies, i.e., biopreservative agents used in various combinations did not annihilate each other. A summarizing of the synergistic effects of applied solutions is presented on Figure 2.

In the case of starter cultures, it often happens that they are used together in mixtures of different strains and/or species. A combination of starter cultures seems to have a greater inhibitory effect on foodborne pathogens and spoilage bacteria, probably due to the rapid decrease in pH value. Fast acidification and growth of starter cultures are desirable to minimize the risk of spoilage and process failure. 

Other types of natural antimicrobials that can be used for biopreservation are non-starter protective cultures, which are microorganism cultures that can inhibit the growth of pathogens or spoilage microbiota without changing the sensory, nutritional, and technological properties of food. The combination of protective cultures exerts its antimicrobial effects in various ways, including by competing for resources and generating antagonistic compounds (e.g., bacteriocins and organic acids).

Currently, the mechanism of action of bacteriocins is primarily envisaged in terms of disrupting the cytoplasmic membrane by causing pores or degrading the cell wall. Yet, the mechanism of action of bacteriocins, particularly those that act against Gram-negative bacteria, is still not fully understood [99]. For example, nisin has a limited antimicrobial effect on Gram-negative bacteria while exhibiting strong antimicrobial activity against an extensive range of Gram-positive foodborne pathogens. A pore is formed by nisin in the membrane of the target cell, resulting in the leakage of small molecules and abrupt cell death [15]. On the other hand, pediocin shows extremely strong inhibition activity against *L. monocytogenes* and acts similarly to nisin but the formation of pores is carried out in a different manner [15]. In the case of enterocins, the matter is more complicated, because different enterocins can belong to completely different classes of bacteriocins. Thus, Liu et al. (2022) [15], for example, found that enterocin AS-48 exhibits its action by weakening the membrane to disrupt the bacterial electron transport system, thereby causing cell death. Concurrently, other enterocins enable certain cations such as Na^+^, K^+^, Li^+^, or H^+^ to pass through the membranes of their targets [15]. Consequently, combining the two bacteriocins could be beneficial since their different modes of action might exhibit synergistic and complementary effects against the target microorganism.

A variety of secondary microbial metabolites with other microbial hurdles were also used as biopreservatives in included studies. Each of them may have a different mechanism of action, and act in combination with another agent to enhance its effect.

In most of the articles, bacterial cultures in the form of starter cultures and non-starter protective cultures were directly incorporated into meat products. Fresh or lyophilized agents were used in either case. Different methods of biopreservative agents’ incorporation into meat products have been employed, ranging from direct addition to fresh meat batter and dipping methods to surface spraying on fresh or ready-to-eat products. Direct incorporation into meat was also used in cases where the antimicrobial effect of purified/semi-purified secondary microbial metabolites was assessed within combined microbial hurdles. Yet another way of applying biopreservatives can be through incorporation in an active packaging structure or microencapsulation but such solutions were not used in the studies included in this review. Nevertheless, such methods could be valuable alternatives to the direct application of biopreservatives to overcome the weaknesses associated with this technique, through preventing degradation during storage and gradual release of the antimicrobial agent, for example. 

Combining the knowledge gathered in the included studies and the available research data, it is concluded that the mixture of microbial hurdles for meat biopreservation may have a potential synergistic effect (Figure 2), mainly through increased antimicrobial activity. The combination of microbial hurdles could induce a stronger antimicrobial effect, expand the spectrum of antibacterial action, and prevent the regrowth of foodborne pathogens and spoilage bacteria. Due to this effect, reduction of the dosage of antimicrobial agents used may be possible, for both microbial or also chemical additives. Fewer additives will reduce the chance of adverse effects such as toxicity or unacceptable sensory properties of meat products caused by their high number or concentration. It will also allow for cost savings in food biopreservation. The approaches presented here thus qualify as an energy-saving and environment-friendly operation.

## 5. Conclusions

The review aimed to find the answers to three questions/problems, which were identified at the beginning of the work. Firstly, a positive effect of using a combination of various bacterial antimicrobials in inhibiting the growth of pathogenic and spoilage bacteria in meat products was shown in the majority of included studies. Secondly, the main mechanisms of action and/or compounds with the potential to be used in biopreservation have not been not clearly identified and require further investigation. Modern proteomic, genomic, and transcriptomic tools may prove to be a great help in discovering the basic mechanisms behind the combined effect of various microbial hurdles. Although the use of various starter or bioprotective cultures in combination is not a new antimicrobial approach, as they have a long history of combined use, it is still necessary to seek to understand the mechanism behind such combined antimicrobial action, because it is not simply the sum of the effects of two separate factors. Thirdly, the application of the discussed solutions in the meat products industry is possible. Biopreservation is one of the most dynamically developing sustainable and promising approaches to enhance food safety, in particular, because it allows for reduction in or elimination of the use of chemical preservatives in food processing. However, the methods and doses of applications should be individually determined. This requires optimization and approvals from the authorities.

To sum up, there has been an increase in consumer demand for “clean label” products over the past few years. Microbial cultures and their bioactive metabolites used in various combinations are promising candidates as green and innovative strategies, providing several benefits—inhibition of food spoilage, minimizing food safety risk, satisfying the consumers, and maintaining economic and environmental resources, that can lead to a sustainable food industry, especially for meat and meat products.

## Figures and Tables

**Figure 1 foods-12-01430-f001:**
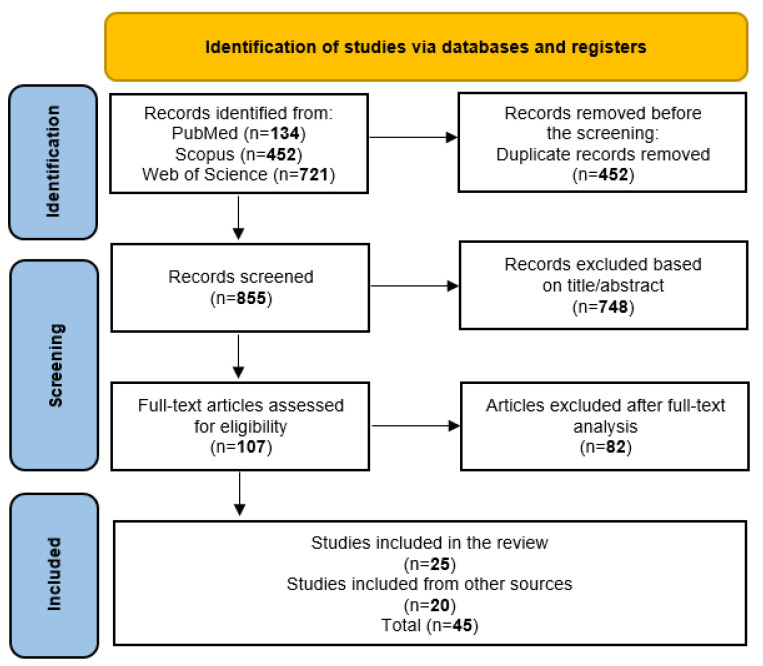
PRISMA flowchart illustrating the identification of studies for inclusion.

**Figure 2 foods-12-01430-f002:**
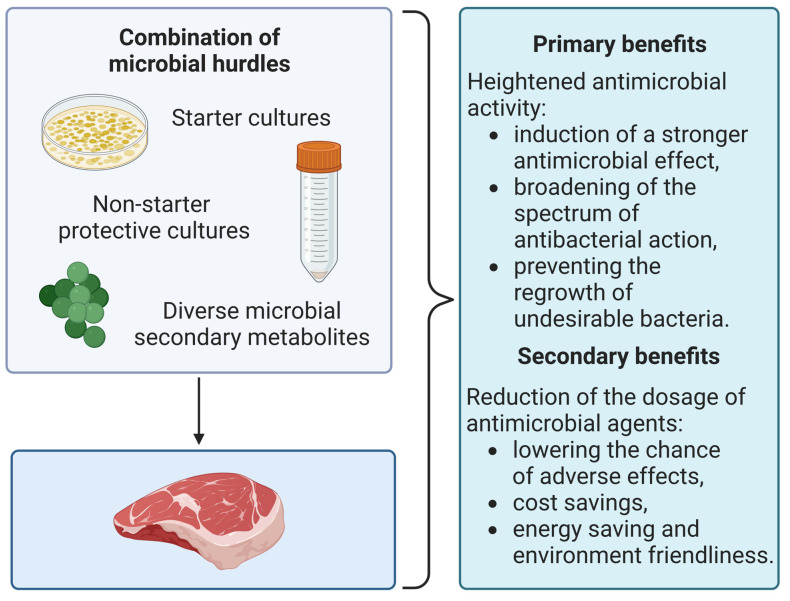
Starter cultures, non-starter protective cultures, or other microbial metabolites can be applied in diverse combinations to meat to gain advantages related mainly to foodborne pathogens and spoilage bacteria inhibition (primary benefit) and lowering the dosage of antimicrobial agents (secondary benefit). Created with BioRender.com (accessed on 27 February 2023).

## Data Availability

Not applicable.

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
