# Peer review of "Synergistic Effect of Combination of Various Microbial Hurdles in the Biopreservation of Meat and Meat Products—Systematic Review"

_foods, 2023, doi:10.3390/foods12071430_

Round 1
Reviewer 1 Report
The review is well conceptualized and presents very useful data for an overview of the different effects of various microbial hurdles in biopreservation in meat industries. It is systematically written and describes their applications and the role of a synergistic effect of various microbial hurdles as biological strategies applied in combination to improve meat safety, especially control of the proliferation or elimination of undesirable microorganisms in different meat products.
The topic is relevant, and it does address a specific gap in the field of biopreservation in meat industries. In this paper, all published works in the field of meat bio-preservation are summarized in a good way.
The authors could perhaps consider in the methodology of use and antimicrobial effects of some essential oils and their active substances (carvacrol, eugenol, etc.) in food biopreservation.
The conclusions are in accordance with the presented arguments, and they do address the main questions posed. The references are appropriate, and the tables are clearly and obviously displayed.
Author Response
Dear Reviewer,
thank you for review and giving us the opportunity to submit a revised draft of the manuscript “Synergistic Effect of Combination of Various Microbial Hurdles in the Biopreservation of Meat and Meat Products – Systematic Review” for publication in the Foods. We appreciate the time and effort that you dedicated to providing feedback on our manuscript and are grateful for the insightful comments on and valuable improvements to our paper. We have incorporated most of the suggestions made by the reviewers. Those changes are matched using the red font function of Microsoft Word. We responded to the reviewers’ comments and concerns. All page numbers refer to the revised manuscript file with tracked changes. In addition, all spelling and grammatical errors found after another reading have been corrected. Please see Report Notes file.
Regarding the similarity rate according to iThenticate report, regrettably, we included a few fragments in the manuscript text, which were also duplicated in the submitted PROCEED-23-00084 study protocol, which significantly affected the total number of repetitions. Accordingly, we have revised the manuscript replacing most of the highlighted fragments from the iThenticate report. Those changes are matched using the yellow underline function of Microsoft Word.
We also have attached the publication license from BioRender.com of Figure 2 to proof of our publication rights.
We look forward to hearing from you to respond to any further questions and comments you may have.
D.Zielinksa and co-authors

Reviewer 2 Report
The review with the title "Synergistic Effect of Combination of Various Microbial Hurdles in the Biopreservation of Meat and Meat Products – Systematic Review" as a part of Meat section (Meat Microflora and the Quality of Meat Products, Special Issue) has 32 pages, 9 Tables, 2 Figures.
The aim of this work was to review the actual knowledge about mechanisms of synergistic effect of various microbial hurdles as biological strategies applied in combination to improve the meat safety, especially for the meat products control.
The review processes the methodological topic, as a review. However, this review needs to be modified in the sense of notes and I give to the authors to take note of my other recommendations.
Keywords: I recommend adding the following words, their number of generally taken criteria for review somewhat small. In addition, they are repeated in the title, allowing keywords to search for databases and at the same time to point out the scope and definition of the topic of the article.
The tables listed in the article exceed the page. Formal adjustment is required and I am not very compliance with the context of the data in the tables.
For example, Table 2: Studies Performed in Meat and Meat Products Testing Diverrse Combinations of Starter Cultures Against Different Target Microorganisms. There are references, but is it not clear why and is this sample of publications comparable? What key has been used?
Is this table to be on 5 site reviews?
The text lacks schemes and wider discussions and conclusions that support it.
Author Response

(The authors gave the same response as above.)

Round 2
Reviewer 2 Report
The review with the title "Synergistic Effect of Combination of Various Microbial Hurdles in the Biopreservation of Meat and Meat Products – Systematic Review"
In the manuscript of the review, the authors made changes and answered on the comments.
Although the article could still be improved, I appreciated the condition of the manuscript that the team improved.
I do not have any specific comments. If other reviewer has no remarks as well as editor of the journal, so it is possible to send the article to proffreading process.